Grazing influences Stipa breviflora seed germination in desert grasslands of the Inner Mongolia Plateau

Liu Wenting
Wang Tianle
Zhang Shuang
Ding Lijun
Wei Zhijun nmndwzj@163.com
College of Grassland, Resources and Environment, Inner Mongolia Agricultural University , Hohhot , China
U’Ren Jana
Electronic publication date: 2018 Mar 1
Publication date: 2018
Volume: 6
Electronic Location ID: e4447
Received 2017 Oct 23; Accepted 2018 Feb 13
Copyright: ©2018 Liu et al.
Copyright year: 2018
Copyright holder: Liu et al.
License: This is an open access article distributed under the terms of the Creative Commons Attribution License, which permits unrestricted use, distribution, reproduction and adaptation in any medium and for any purpose provided that it is properly attributed. For attribution, the original author(s), title, publication source (PeerJ) and either DOI or URL of the article must be cited.
License URL: https://creativecommons.org/licenses/by/4.0/

Keywords: Reproductive strategy, Seed number, Seed phenotypic characteristics

Funding: National Natural Science Foundation of China 31460126 National Key Basic Research Program of China 2014CB138805 This work was supported by the National Natural Science Foundation of China (No. 31460126), and the National Key Basic Research Program of China (No. 2014CB138805). The funders had no role in study design, data collection and analysis, decision to publish, or preparation of the manuscript.

==============================
Seed germination plays an important role in determining the composition and regeneration of plant populations (Stipa breviflora). However, the influencing factors and strategies employed for seed germination in desert grasslands under grazing remain unknown. Therefore, in this study, the reproductive allocation, seed density, seed properties, and corresponding seed germination rates of S. breviflora were examined. Possible situations encountered during dispersal were also simulated to determine their effects on seed germination. The results showed that reproductive individual density not subjected to grazing were significantly higher than those subjected to moderate and heavy grazing. For seed density and seed bank in soil, the highest values were observed for the no grazing treatment, followed by the moderate and heavy grazing treatments. The seed density for germination of soil seed banks was nearly one-fourth of seed density during the growing season. In addition, grazing treatments affected the phenotypic characteristics of seeds and reduced the lower limit of the weight of germinable seeds. Awn removal significantly increased germination. The longest germination time was observed for seeds that entered the soil at an angle of 0°. Our research demonstrated that grazing negatively affected the desert grassland edificator. Individual plants adopted different adaptation strategies under different grazing intensities; for example, a fixed proportion of the seed number and seed germination number of S. breviflora in the soil seed bank was maintained by exceeding the minimum weight of a seed for seed germination. During seed dispersion, the awn effectively prevented germination under unfavourable conditions and helped seeds enter the soil at an optimal angle for promoting germination.

Introduction

As a critical and the most basic process for completing the life cycle of a plant, propagation affects plant population dynamics, ensures population stability, and maintains grassland biodiversity and ecosystem health. The success of seed germination determines the reproduction and survival of plant populations (Rajjou et al., 2012) and directly affects the time a plant enters an ecosystem and begins the accumulation of dry matter (Weitbrecht, Müller & Leubner-Metzger, 2011). Therefore, seed germination is both economically and ecologically important.

The potential mechanisms of seed germination have been analysed with respect to temperature (Khan & Ungar, 1998), light (Yamaguchi & Kamiya, 2001), moisture (Foolad, Zhang & Subbiah, 2003), soil acidity and salinity (Khan & Ungar, 1997), chemical substances and burial depth (Okamoto et al., 2010; Hu, Wu & Wang, 2009). Although these studies have greatly enriched our understanding of factors that influence seed germination, they were performed under limited and controllable experimental conditions. However, in the wild state, the propagation process of plants is frequently threatened by the risk of falling prey to herbivores in addition to the abovementioned physical threats (Yagihashi, Hayashida & Miyamoto, 1998).

Severe disturbance caused by herbivores, such as the repeated feeding of livestock, destroys the vegetative and reproductive organs of plants and causes negative effects such as shortened and narrowed leaves and reduced inflorescences. The theory of life history suggests that the distribution of plant resources between growth and propagation is highly flexible (Miller, Tyre & Louda, 2006). To compensate for the damage caused by grazing, when the total amount of available resources is constant, the plant uses more resources to produce vegetative organs and sacrifices sexual reproduction. With continued grazing pressure, the amount of plant production will be reduced, causing a reduction in the soil seed bank. In this scenario, what types of ecological strategies will seeds adopt for adaptation? Louault et al. (2005) compared properties of grassland plants under 12 years of continuous grazing and non-grazing treatments and found that grazing also leads to a chain of effects such as a narrowing of seed shape and a reduction in seed weight.

Seed size directly affects the propagation, spread and survival probability of seeds (Chambers, 1995). Compared with large seeds, small seeds can spread farther with the aid of an awn, thereby improving their survival fitness (Coomes & Grubb, 2003). For Stipa, widespread on the Eurasian steppe, the awn helps spread seeds efficiently. For example, the pappus of the awn is erect when the air is dry to increase air buoyancy and reduce the seed-dropping rate (Peart, 1984; Greene & Johnson, 1993). The pappus can also stick to the fur of animals to disperse. The direction of seed drop greatly affects seed anchoring in soil, and the moisture-absorbing effect of the awn can increase the self-embedding line of the seed (Peart, 1981; Ghermandi, 1995). Thus, we propose that the angle at which a seed enters the soil affects the germination of that seed. The abovementioned studies illustrate the importance of the awn to the seed, but the plant’s reproductive organ will inevitably be injured by livestock feeding, trampling, etc (Hickman & Hartnett, 2002). In addition, what type of strategy will a seed adopt after the loss of the awn?

A large number of studies have focused on plant reproduction allocation or the response of seed germination to changes in the germination environment. However, we supposed that the seed not only is a product of the tradeoff associated with the reproductive allocation of plants but also affects seed propagation and germination under grazing. To the best of our knowledge, such research has not yet been reported. In this study, the following aspects of seed germination were examined: (1) the reproductive strategy of a perennial grass (Stipa breviflora) under different grazing treatments and the effect of seed phenotypic characteristics on seed germination under a particular allocation strategy; and (2) the effect of the awn column on seed germination characteristics and the effect of the seed implantation angle on seed germination characteristics following seed scattering to soil.

Methods

Study area

The study area is in Zhurihe (112°47′16.9″E, 42°16′26.2″N) at an elevation of 1,100–1,150 m at the S. breviflora grassland station of Inner Mongolian University on the Inner Mongolian Plateau. Rain and high temperatures occur over the same period at the site, which has average annual rainfall of 183.0 mm, an annual average temperature of 5.8 °C, an annual sunshine duration of 3137.3 h, annual evaporation of 2,793.4 mm, and a frost-free period of 177 days, together composing a moderate temperate climate. Furthermore, the average wind speed is 5.1 m/s, with winds mostly concentrated in winter and spring and dominated by winds from the northwest. The average number of windy days is 67, and sandstorms do not occur. The soil is a chestnut soil, with surface desertification and with a humus horizon measuring 20–30 cm thick. The vegetation layer is short and low, with a typical height of 10–25 cm; the vegetation coverage is also low, approximately 15–25%. Perennial grasses, particularly S. breviflora, are the primary dominant plants.

Experimental design

The experimental treatments for this study were applied in a continuous section with flat terrain and a relatively uniform environment; therefore, differences in background and spatial heterogeneity were effectively controlled. The grazing experiment began in 2010, and grazing began in May and ended at the end of October each year. A continuous grazing method was adopted. In this study, three grazing treatments were applied: heavy, moderate and no grazing, with three replicates per treatment. Each treatment area was 2.6 ha. The stocking rates for moderate and heavy grazing were 1.92 and 3.08 sheep ha−1 a−1, respectively. The grazing herbivores were “Sunite sheep”. The health status, individual size, weight and sex of the sheep in the grazing areas were essentially the same.

Data collection

On June 1, 2015, field sampling was conducted. In each treatment, three randomly located 1 × 1 m quadrats were sampled. In each quadrat, the numbers of S. breviflora reproductive and non-reproductive individuals were recorded, and then each S. breviflora was clipped at ground level and numbered. The fresh plant material was placed in an oven at 105 °C for 10 min to de-enzyme and then oven-dried at 65 °C for 48 h and weighed. For reproductive individuals, the reproductive and foliage branches were separated and weighed. In addition, the percentage of the productive branch and the percentage of the foliage branch were calculated. Subsequently, manual threshing was conducted, and the number of seeds produced by the reproductive individuals in each quadrat was counted.

Soil seed bank samples were collected in the middle of the quadrat, with the length, width and depth of the samples being 10, 10 and 5 cm, respectively, and then the soil samples were packed and transported to the laboratory. For the determination of the soil seed bank, the seed germination method was used to estimate the number of S. breviflora seeds in the soil seed bank. On June 20, 2016, the germination experiments were performed in the glasshouse (the temperature during the germination experiment (Jun.–Sept.) was 24.4–26.8 °C) of the Department of Agrostology of Inner Mongolia Agricultural University. Gravel, rhizomes, and fallen leaves in the same layer as the soil samples at the same sampling site were removed, and the samples were evenly mixed. A germination box measuring 15 × 15  × 7 cm and lined with a 1-cm-thick vermiculite layer was used, and this layer was then covered with the soil samples collected from the field. Water was regularly supplied to the water sink to maintain soil moisture. The number of S. breviflora seeds that germinated in each germination box was recorded. Germination continued until August, and the experiment was terminated after an additional month of observation.

On June 3, 2015, the mature reproductive branches of healthy S. breviflora plants with a high seeding rate were selected in each experimental area and manually threshed after drying; then, seeds were stored in a bag over the winter for preservation. On June 15, 2016, these seeds were returned to the laboratory, evenly mixed with those from the same grazing treatment and divided into three portions:

(1) In the first portion, 50 seeds from each grazing treatment were randomly selected, and the seed width, length, and weight and the awn weight were measured with a Vernier calliper (Fig. 1). After the morphological indices were measured, each seed was rinsed with tap water, disinfected with 75% ethanol, rinsed with deionized water, and then placed in a petri dish. The seed germination of S. breviflora in the petri dishes was recorded every 24 h. After germination was recorded, an appropriate volume of water was added to ensure the same treatment conditions in a later part of the study. The cultivation was terminated after 14 continuous days.

(2) In the second portion, based on the seeds in the first portion, seeds with good morphological characteristics and a minimum germinated seed weight were selected, and the awn was removed or left intact (Fig. 2). These seeds were placed in petri dishes, with 50 seeds per dish, and each treatment had five replicates. Then, the germination process in (1) was repeated.

(3) In the third portion, seed selection was repeated and the awn was removed. These seeds were planted in the soil at soil surface angles of 0°, 15°, 30°, 45°, 60° and 90°; 3/4 of the volume of each seed was in the soil. Each grazing treatment was replicated 10 times, with moderate moisture, and then the germination process in (1) was repeated (Fig. 3).

Figure 1 Seed morphological characteristics. (A) seed length; (B) seed width; (C) awn.

Image by Guopeng Liu.

Figure 2 Seed germination experiment. (A) seed with awn unremoved; (B) seed with awn removed.

Image by Guopeng Liu.

Figure 3 Germination experiment of seeds entering soil at different angles.

Image by Guopeng Liu.

Statistical analyses

One-way ANOVA was used to analyse the densities of non-reproductive and reproductive individuals; the percentage of reproductive branches; the percentage of foliage branches; seed density; the soil seed bank; the width, length and weight of seeds; the awn weight; seed germination with or without an awn; and the differences in the number of S. breviflora germinated in soil from different angles under the three grazing treatments of heavy grazing (HG), moderate grazing (MG) and no grazing (NG). Multiple comparisons were performed with Duncan’s tests. A two-way ANOVA was conducted to assess the effects of grazing treatment and seed awn or seed soil entry angle on seed germination. Linear regression analysis was conducted on the seed germination days and seed soil entry angles. The abovementioned statistical analyses were performed with the SPSS 19.0 (SPSS, Inc., Chicago, IL, USA) statistical software package.

Results

Distribution of reproductive individuals and seed density

For S. breviflora under different grazing treatments (Figs. 4A–4D), at a sample scale of 1 × 1 m, the reproductive and non-reproductive individual density were significantly higher in the no grazing treatment than those under moderate grazing (P < 0.05), with approximately 3-fold more reproductive individuals under the no grazing treatment than under grazing; values were not significantly different between moderate and heavy grazing. At the scale of the reproductive individual, the results showed that the weight of reproductive branches of S. breviflora under the moderate grazing treatment was significantly higher than that under the no and heavy grazing treatments (P < 0.05), with an increase of approximately 82.99%. The highest seed density of S. breviflora occurred under the no grazing treatment, followed by moderate grazing and then heavy grazing (Fig. 4E), while the difference in seed density of S. breviflora in the soil seed bank between moderate grazing and heavy grazing was not significant (Fig. 4F).

Figure 4 Characteristics of Stipa breviflora (mean ± standard error) under heavy grazing, moderate grazing and no grazing.

Different letters indicate significant differences at P < 0.05.

Phenotypic characteristics and seed germination

The results of one-way ANOVA (Figs. 4G–4J) showed that grazing livestock significantly reduced seed length, width, and weight and awn weight, with values significantly higher in the no grazing treatment than in the moderate and heavy grazing treatments. However, no significant difference was detected between the moderate and heavy grazing treatments. The minimum weight of germinated seeds (Fig. 5) under no grazing exceeded 1.20 mg, with an average weight of 1.80 mg, which was 37.07% higher than that under grazing. The minimum weight of germinable seeds under heavy grazing exceeded 0.90 mg.

Figure 5 Germinable seed weight under heavy grazing (HG), moderate grazing (MG) and no grazing (NG).

Seed germination with awn removed or not removed

To further explain the seed germination characteristics, and based on the good morphological characteristics and minimum weight of germinated seeds, we selected seeds under relatively ideal conditions for germination to perform awn-removal and non-removal experiments (Fig. 6). The seeds began to germinate at day 3, and germination peaked on days 3–5 and then stabilized. In the awn-removal treatment, the seed germination rate under the moderate grazing treatment reached 83.60%, which was 49.29% higher than that of non-removal and nine-fold higher than that of non-removal under the no grazing treatment. Additionally, based on the results of the two-factor ANOVA (Table 1), the germination rate of S. breviflora was affected by grazing treatment (Table S1) and awn-removal treatment, but no statistical interaction was observed (Table 1).

Figure 6 Germination characteristics of seeds under heavy grazing, moderate grazing and no grazing.

(A–C), seed with awn unremoved; (D–F), seed with awn removed. (A) and (D) indicate heavy grazing; (B) and (E) indicate moderate grazing; (C) and (F) indicate no grazing. Error bar means standard error.

Table 1 Two-way ANOVA of seed germination.

Effect	Degrees of freedom	F value	P value	
Grazing	2	34.01	<0.0001	
Seed awn	1	32.62	<0.0001	
Grazing × Seed awn	2	2.49	0.1038	
Grazing	2	13.66	<0.0001	
Seed entering soil at different angles	5	5.68	<0.0001	
Grazing × Seed entering soil at different angles	10	0.39	0.951	

Angle of seeds entering soil and germination

The results (Fig. 7) of the germination experiments regarding the angles of seeds entering soil showed that for the seeds that entered soil at an angle of 0°, the average germination time exceeded nine days. The number of germination days of S. breviflora seeds entering the soil at different angles fit the relationship described by a quadratic equation (y = ax2 + bx + c). The two-factor ANOVA showed that the angle of S. breviflora seeds entering the soil significantly affected the germination days (P < 0.0001; Table 1).

Figure 7 Relationships between seed germination days and seed entering soil angles under heavy grazing, moderate grazing and no grazing.

(A) indicates heavy grazing; (B) indicates moderate grazing; (C) indicates no grazing. Error bar means standard error.

Discussion

Shroff et al. (2008) argued that selective feeding severely injures plants because the most attractive part of a plant to the animal is also the most reproductively valuable part of the plant (fruits and seeds, among others) (Gómez et al., 2008), which is consistent with the results of this study (Fig. 4). The numbers of seeds demonstrated that grazing had a negative effect on desert grassland vegetation (Koerner et al., 2014) and suggest that the feeding selection habits of livestock in habitats are one of the contributing factors in the partial extinction of certain grassland species; moreover, when these species are not supplemented or effectively renewed, the risk of such extinction can be high. Furthermore, the number of seeds for germination in the soil seed bank of S. breviflora were nearly one-fourth the number of seeds collected, which indicated that the seed-to-seedling stage is usually the time when the rate of return on investment of plants is relatively low (Rajjou et al., 2012), indirectly demonstrating the grazing resistance characteristic of perennial grasses in a desert grassland.

In areas with soil depletion and scarce rainfall, such as desert grasslands, the propagation strategy of a plant is the key to the extension and continuous expansion of the population (Wang et al., 2017). In the present study, grazing treatment reduced the density of reproductive individuals in the S. breviflora population (Fig. 4), which to some extent verifies the hypothesis that grazing negatively affects plants. However, the weight of reproductive branches was significantly higher under moderate grazing than under the no grazing and heavy grazing treatments (P < 0.05). Although the results appear contradictory, they might be explained by the following factors. The first is the calculation method. In ecology, the computation of plant population density is generally based on counts of individuals within a certain area (Tian et al., 2017), and this population density is assumed to represent the average state of the individuals of the population within the sampling area. However, surviving in the same state of subsistence is nearly impossible for individual plants; thus, the characterization of plant populations should fully account for the internal structure of the population. The second is the grazing livestock. Previous studies have shown that S. breviflora is not a preferred species for sheep (Liu et al., 2016). Under moderate grazing, various grassland species were available, which provided sheep with more grazing options (Provenza, 1995) and opportunities to graze a greater variety of species more frequently; thus, S. breviflora had adequate living conditions and internally adjusted its resource allocation. However, when food was relatively limited (i.e., heavy grazing), the animals had relatively few feeding options and aimed only to fill their stomach. This behaviour was an indirect indication that the selection of food by animals was not only limited to their own preferences or maximum energy intake but also adjusted to meet their nutritional requirements (Simpson et al., 2004). Therefore, because the sheep switched from actively selecting food to passive, random selection, S. breviflora was required to adopt the strategy of reducing its numbers of individuals to avoid being grazed, which led to the chain reaction involving a decrease in seed number.

With the reduction in seed density, the seed morphological characteristics were also affected, and grazing significantly reduced the seed length, width, and weight and the awn weight, which further validated the hypothesis that grazing negatively affects plants. Additionally, the average weight of germinable seeds in the no grazing treatment was 37.07% higher than that in the grazing treatments (Fig. 5), with the lowest value being 1.33-fold higher than that of heavy grazing. This result indicates that grazing breaks and reduces the threshold for seed germination, which might be an effective strategy for plants under the long-term grazing pressure of herbivores. Hughes et al. (1994) found that seeds that weighed more than 100 mg generally spread with the aid of vertebrates, whereas those that weighed less than 0.1 mg tended to use gravity for dispersal. For those seeds with weights between these two values, a variety of dispersal mechanisms were observed. The seeds we studied all had a weight between these two weights (Fig. 4), which suggested that the seeds under grazing could more effectively spread and colonize than those under no grazing because, with the same shape and structure at the same wind speed, smaller seeds spread farther (Matlack, 1987), with the additional help of grazing livestock. Therefore, this strategy may be effective for plants to cope with herbivorous behaviour.

Many studies suggest that rapid seed germination is a common survival mechanism adopted by plants in arid areas (Wallace, Rhods & Frolich, 1968); however, our results are not fully consistent with this generalization. In this study, the germination peak appeared from day 3 to 5 and then steadily stabilized (Fig. 6), which might be a response to the scarce rainfall in desert grasslands. Although sufficient rainfall can stimulate seed germination, S. breviflora might adopt an incessant germination strategy in which not all seeds germinate in response to a single event. If plant seeds are fully responsive to the first rainfall event, leading to all seeds germinating simultaneously, and then a long-term drought occurs, in which all seedlings die, the population will be at risk of extinction because the seed bank is exhausted. Additionally, based on field observations, a considerable portion of seeds were hung by the awn on the leaves of S. breviflora or stuck in short bunches, which we also interpreted as germination strategies that will ensure continuous seed germination while also delaying seed germination and reducing the risk of seed bank exhaustion.

Our study showed that awn removal significantly increased the seed germination rate (Fig. 6), indicating that awns delayed seed germination and, to a certain extent, providing further evidence that S. breviflora adopted a “prudent” germination strategy. Further study revealed that the angle at which the seed enters the soil significantly affected seed germination (Table 1). The number of seed germination days was consistent with the equation based on the fitted relationship (P < 0.05; Fig. 7), which demonstrated that the falling angle of a propagule and the self-burial behaviour of the seed markedly affected seed germination. Based on the abovementioned experiments, we believe that the awn of the propagules most likely plays a highly important role in plant propagation. With the maturation of the seed, the propagule falls from the spikelet with the aid of awn twisting (Raju & Ramaswamy, 1983) and disperses the population through transport using a medium such as wind or animals. During the dispersal process, the awn column effectively suppresses seed germination when in an unfavourable condition; however, when the seed falls to the ground, the awn aids entrance into the soil through hygroscopic absorption and then breaks away from the seed at a joint to avoid predation by cereal-eating animals.

Conclusion

In conclusion, our results demonstrated that grazing had negative effects on desert grassland plants, directly reducing the number of reproductive individuals and the phenotypic characteristics of seeds (seed length, width, and weight, and awn weight). Among individuals, desert steppe plants maintained a fixed ratio of seeds of the soil seed bank to the seeds of plant production by reducing the lower limit of the minimum weight of germinable seeds. During seed dispersal, the awn effectively prevented the germination of seeds in unfavourable conditions and helped the seeds enter the soil at a certain angle to promote germination.

Supplemental Information

Table S1 Seed germination (mean ± standard error) under heavy grazing (HG), moderate grazing (MG) and no grazing treatments (NG)

Different letters indicate significant differences at P < 0.05.

Click here for additional data file.

Data S1 Raw data

Click here for additional data file.

We thank Guopeng Liu for his detailed graph (Figs. 1–3). We also gratefully acknowledge students from Inner Mongolia Agriculture University for their help with fieldwork.

Additional Information and Declarations

Competing Interests

Author Contributions

Data Availability

The authors declare there are no competing interests.

Wenting Liu conceived and designed the experiments, performed the experiments, analyzed the data, prepared figures and/or tables, authored or reviewed drafts of the paper, approved the final draft.

Tianle Wang, Shuang Zhang and Lijun Ding performed the experiments, contributed reagents/materials/analysis tools, authored or reviewed drafts of the paper, approved the final draft.

Zhijun Wei conceived and designed the experiments, performed the experiments, authored or reviewed drafts of the paper, approved the final draft.

The following information was supplied regarding data availability:

The raw data have been provided in a Supplemental File.

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
