# Peer review of "Grazing influences Stipa breviflora seed germination in desert grasslands of the Inner Mongolia Plateau"

_PeerJ, doi:10.7717/peerj.4447_

## Round 0.1 · original submission · Major Revisions

The reviewers commented positively on the manuscript; however, there are a number of areas where the manuscript can be improved. Importantly, the English language needs of revision to clarify what the authors mean to say, the introduction needs additional information on how this study fills a specific knowledge gap, and more information is needed to clarify the methodology.

Reviewer 1 ·

Basic reporting

no comment

Experimental design

no comment

Validity of the findings

no comment

Additional comments

This study is exploring the influencing factors and strategies of seed germination in desert grasslands under grazing treatment. It is very important to know the reproductive allocation, seed density, seed properties, and the corresponding seed germination rates of a dominant species for restoration of disturbed grassland. I think this study is well designed and paper is well written by authors whose English is the second language. I just have several small questions.
1: Please add full genus name for “S. breviflora” in abstract.
2: The size of each treatment?
3: There are too many figures. I think you can combine several figures. As for tables, it is the same problem.

Reviewer 2 ·

Basic reporting

This manuscript did a nice job of reviewing relevant literature in the introduction to give the context of the work. The structure conforms to PeerJ standards, the figures are relevant and well described, and the raw data are supplied.

However, there are many places where the English language is unclear. For instance, Lines 187-190, the meaning of this sentence is not clear to me as a reader, in order for the findings to be accessible to an international audience the language would need to be clarified here and elsewhere throughout the manuscript.

Experimental design

The experiments and observations represent original primary research within scope of the journal. However, the case could be stronger for how this fills a knowledge gap in the literature.

Within the methods, there were a few places where there was not enough justification for the chosen method, or not enough detail to replicate the measurements:

Line 125, is 14 days long enough to survey germination in this species?

Line 162, it's interesting that you're reporting the "weight exceeded by germinable seeds", do you mean this is the minimum weight of a seed that will germinate? This needs to be defined more clearly.

Line 166 - you state that you chose "close to ideal" seeds for the awn removal experiment, what characteristics did you use to identify those ideal seeds?

Aspects of the statistical analysis should also be made clearer. For instance: Line 170 "Additionally, based on the results of the two-factor ANOVA (Table 1), the germination rate of S. breviflora was affected by grazing treatment and awn-removal treatment." Looking at the Table, I see that both grazing treatment and awn removal are significant as main factors, but there is not a statistical interaction. To be clear on the findings, this should be stated in the results. Also, you should state how grazing treatment influenced germination rate, did grazing result in a decline in germination rate?

Validity of the findings

The Discussion is well structured around the primary findings and cites relevant literature to place the results in context. However, the Conclusions paragraph is confusing. For instance, lines 253-255, " Among individuals, desert steppe plants maintained a fixed proportion of
seeds for germination in the soil seed bank for the number of plant species by reducing the lower limit of seed germination potential." Since this experiment was focused on just one species, the "number of plant species" seems incorrect here. Also "reducing the lower limit of seed germination potential" is also unclear. I think this conclusion means that under grazing there are similar numbers of seeds in the seedbank per reproductive individual, but the seeds are smaller.

Minor:

Lines 182-184, these data should be presented in the Results section.

Additional comments

In summary, the manuscript is well written and carefully prepared. However, the English language requires revision to make all statements of findings clear. The introduction could also be improved to clarify why this study fills a knowledge gap.

Reviewer 3 ·

Basic reporting

1) the language needs further improvement.
2) some background information of sub-experiment needs to be provide.

Experimental design

Some methods for parameter calculation, such as, weight ratio of productive branch, need to supply.

Validity of the findings

Some conclusions are over speculated.

Additional comments

General Comments:
Liu et al. reported the results of the effects of grazing intensity on seed germination characteristics of Stipa breviflora by a field grazing experiment in a desert steppe of Inner Mongolian grassland. Also, they examined the impacts of awn presence or absence and the angles of seeds entering soil on the germination of the seeds of this species. They found that, the abundance of reproductive individuals as well as seed density and soil seed bank were higher under no grazing treatment than those under mediate and heavy grazing treatments. Awn removal enhanced the germination of the seeds and the angles of seeds entering the soil also exerted strong impacts on seed germination. They concluded that grazing negatively affected grassland vegetation and plants adopted different strategies under different intensities of grazing. This study provides some important information on our understanding of responses of native grassland species to grazing intensities. However, some issues constrain my recommendation for publishing on PeerJ.
First, the title is not appropriate in my opinion. Influencing factors and strategies are too broad. You should focus on your studied issues. More important, this study only examined the responses of one species, i.e., Stipa breviflora. Other species you did not addressed. But your title did not reflect this feature of your study.
Second, the reproduction process and the germination process are totally different stages though they are successively connected. The mechanisms for species’ responses to these processes may be totally different, therefore, I suggest that you’d better to separate this MS into two ones. One focuses on the reproductive features and the other on the germination of seeds.
Third, in the introduction section, you should present some background of your sub-experiment, for example, why you do the experiment of awn removal. For Stipa breviflora, awn is an important organ for seeds burying themselves as it usually helps the seeds entering the soil. It has been suggested that this organ will not depart from the seeds until the seeds reach the optimal depth. So why do you remove the awn in your experiment? Dose this have general scientific significance?
Finally, some conclusions are over explained. For instance, you stated that “grazing negatively affected grassland vegetation” in your abstract. However, you only examined the response of one species, it can not represent the vegetation response of your community. Intermediate disturbance hypothesis has suggested that ecosystem biodiversity is highest at mediate disturbance intensity and grazing optimization hypothesis suggests that productivity of grassland vegetation reaches a peak at moderate grazing intensity.

Minor Points:
In abstract section, you need not present all results of your experiment, Please focus on the very important findings and the major mechanisms for these findings.
Line 22: add the name of the species after population.
Lines 25-26: the meaning of the sentence ‘The highest … … then heavy grazing’ is unclear. Please replace it with “For seed density and seed bank in soil, the highest values were observed at no grazing treatment, followed by moderate and heavy grazing treatment”
Lines 26-27: the meaning of “The number of seed germination … … 1/4 of the number of seeds” is unclear. Please replace it with “Generally about one quarter seeds in the seed bank germinated during the growing season”.
Many sentences need rewrite for clarity.
Line 59: replace “protection” with “adaptation”.
Line 82: add “annual” before “sunshine”.
Line 88: the information about the community need to supply, as interactions among plant species may affect species reproduction.
Lines 158-159: delete “seed phenotypic characteristics”, this is the results please present the specific parameters you measured.
Line 199: the meaning of “ the perspective of the observations must be considered” is unclear.
Line 202: the meaning of “ the parameters of grazing must be analyzed” is unclear.
Line 230: please give a definition for “prudent germination strategy”.
Figure 4: Please describe the methods for calculating the parameters. For example, how do you estimate weight ratio of productive branch, and of foliage branch.
Figure 6: the unit “g” for seed weight and for awn weight is not appropriate, “mg” or “µg” may be more suitable. Similar issue exists in Figure 7.

---

## Round 0.2 · Minor Revisions

Thank you for your careful response to the suggestions made by the previous reviewers. As a result of these changes, I think that the manuscript has been substantially improved and is suitable for publication pending minor edits to address the points raised by the reviewer, including adjusting the title as suggested. However, please note that I do not feel you need to rewrite the discussion section.

Reviewer 1 ·

Basic reporting

The English writing is good. But some places is not professional.

Experimental design

The experimental design is reasonable. Authors provide enough details.

Validity of the findings

Well done.

Additional comments

The manuscript is edited very well after authors completed the first revision. I suggest acceptance if authors do some minor revision.

Title: Strategies of Stipa breviflora seed germination under grazing: a study of the desert grassland edificatory on the Inner Mongolia Plateau
In this study, I think you mainly explore whether or not seed germination was influenced by grazing. So I suggest you can change the title to “Grazing influences Stipa breviflora seed germination in desert grasslands of the Inner Mongolia Plateau”

Line 24 “reproductive individuals not subjected to grazing were significantly more abundant”, what does the “abundant” mean here?

Line 26-27 “Generally, approximately one-quarter of the seeds in the seed bank germinated during the growing season.” It does not sound like scientific writing.

Line 30-31 “Our research demonstrated that grazing negatively affected grassland vegetation.” Here authors just test one species, so the conclusion is not reasonable.

Line 29 change “dramatically” to significantly.

In discussion part:
In the first paragraph, if you write down clearly the most important findings of your study, it would be easier to understand your study for readers.
For each paragraph, what you mainly want to say, please write in the first sentence or the last sentence.

---

## Round 0.3 · accepted · Accept

Thank you for addressing the points raised by the reviewers. I believe the result is a stronger paper and I am pleased to accept it for publication.